# Supervising Unsupervised Learning

**Vikas K. Garg**
CSAIL, MIT
vgarg@csail.mit.edu

**Adam Kalai**
Microsoft Research
noreply@microsoft.com

## Abstract

We introduce a framework to transfer knowledge acquired from a repository of (heterogeneous) supervised datasets to new unsupervised datasets. Our perspective avoids the subjectivity inherent in unsupervised learning by reducing it to supervised learning, and provides a principled way to evaluate unsupervised algorithms. We demonstrate the versatility of our framework via rigorous agnostic bounds on a variety of unsupervised problems. In the context of clustering, our approach helps choose the number of clusters and the clustering algorithm, remove the outliers, and provably circumvent Kleinberg's impossibility result. Experiments across hundreds of problems demonstrate improvements in performance on unsupervised data with simple algorithms despite the fact our problems come from heterogeneous domains. Additionally, our framework lets us leverage deep networks to learn common features across many small datasets, and perform zero shot learning.

## 1 Introduction

Unsupervised Learning (UL) is an elusive branch of Machine Learning (ML), including problems such as clustering and manifold learning, that seeks to identify structure among unlabeled data. UL is notoriously hard to evaluate and inherently undefinable. To illustrate this point, we consider clustering the points on the line in Figure 1. One can easily justify 2, 3, or 4 clusters. As Kleinberg argues [1], it is impossible to give an axiomatically consistent definition of the "right" clustering. However, now suppose that one can access a bank of prior clustering problems, drawn from the same distribution as the current problem at hand, but for which ground-truth labels are available. In this example, evidence may favor two clusters since the unlabeled data closely resembles two of the three 1-dimensional clustering problems, and all the clusterings share the common property of roughly equal size clusters. Given sufficiently many problems in high dimensions, one can learn to extract features of the data common across problems to improve clustering.

We model UL problems as representative samples from a meta-distribution, and offer a solution using an annotated collection of prior datasets. Specifically, we propose a *meta-unsupervised-learning* (MUL) framework that, by considering a distribution over unsupervised problems, reduces UL to Supervised Learning (SL). Going beyond transfer learning, semi-supervised learning, and domain adaptation [2, 3, 4, 5, 6, 7, 8] where problems have the same dimensionality or at least the same type of data (text, images, etc.), our framework can be used to improve UL performance for problems of different representations and from different domains. While the apparent distinctions between the domains might make our view seem outlandish, they are merely an artifact of human perception or representation. Fundamentally, we may encode any kind of data as a program or a bit stream. Depending on considerations such as the amount of available data, one can choose to have a distribution over only the programs from a single domain such as images, and consider all other programs as a set of measure 0; or alternatively, as we do, consider a distribution over all programs.

Empirically, we train meta-algorithms on the repository of classification problems from `openml.org` that has a variety of datasets spanning domains such as NLP, computer vision, and bioinformatics. Ignoring labels, each dataset can be viewed as a UL problem. We take a data-driven approach to

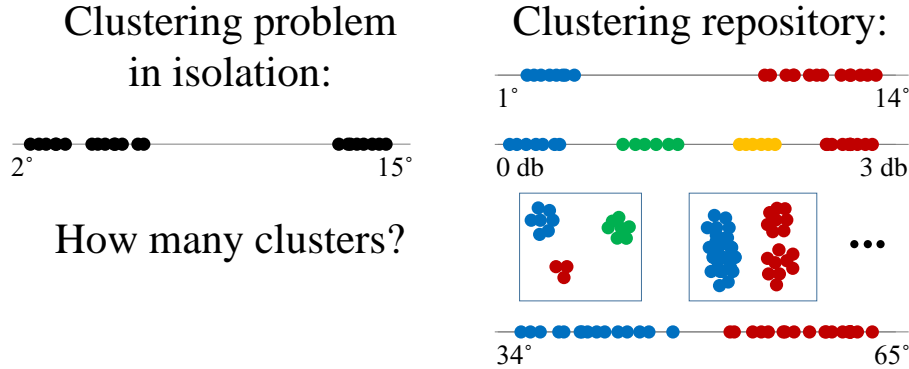

Figure 1: In isolation (left), there is no basis to choose a clustering – even points on a line could be clustered in 2-4 clusters. A repository of clustering problems with ground-truth labels (right) can inform the choice amongst clusterings or even offer common features for richer data.

quantitatively evaluate and design new UL algorithms, and merge classic ideas like UL and domain adaptation in a simple way. This enables us to make UL problems, such as clustering, well-defined. Our model requires a *loss* function measuring the quality of a solution with respect to some conceptual *ground truth*. Note that we require the ground truth labels for a training repository but not test data. We assume access to a repository of datasets annotated with ground truth and drawn from a *meta-distribution* $\mu$ over problems, and that the given data $X$ was drawn from this same distribution (though without labels). From this collection, one could learn which clustering algorithm works best, or, even better, which algorithm works best for which type of data. The same *meta*-approach could help in selecting how many clusters to have, which outliers to remove, etc.

Our theoretical model is a meta-application of Agnostic Learning [9] where we treat each entire labeled problem analogous to a training example in a supervised learning task. We show how one can provably learn to perform UL as well as the best algorithm in certain classes of algorithms. In independent work, Balcan et al [10] design near-optimal algorithms for NP-hard problems such as meta-clustering. Our work also relates to supervised learning questions of meta-learning, sometimes referred to as auto-ml [e.g. 11, 12], learning to learn [e.g. 13], Bayesian optimization [e.g. 14] and lifelong learning [e.g. 15, 16]. In the case of supervised learning, where accuracy is easy to evaluate, meta-learning enables algorithms to achieve accuracy more quickly with less data.

**Our contributions.** We make several fundamental contributions. First, we show how to adapt knowledge acquired from a repository of small datasets that come from different domains, to new unsupervised datasets. Our framework removes the subjectivity due to *rules of thumb* or educated guesses, and provides an objective evaluation methodology for UL. We introduce algorithms for various problems including choosing a clustering algorithm and the number of clusters, learning common features, and removing outliers in a principled way. Next, we add a fresh perspective to the debate triggered by Kleinberg on what characterizes right clustering [1, 17, 18], by introducing the *meta-scale-invariance* property that learns a scale across clustering problems and provably makes good clustering possible. Finally, we use deep learning to automate learning of features across small problems from different domains and of very different natures. We show that these seemingly unrelated problems can be leveraged to gain improvements in the average performance across *previously unseen* UL datasets. In this way, we effectively unite many heterogeneous "small data" into sufficient "big data" to benefit from a single neural network. In principle, our approach may be combined with compression methods for deep nets [19, 20, 21] to learn under resource constrained settings [22], where maintaining a separate deep net for each dataset is simply impracticable.

## 2   Setting

We first define agnostic learning in general, and then define meta-learning as a special case where the examples are problems themselves. A learning task consists of a universe $\mathcal{X}$, labels $\mathcal{Y}$ and a bounded *loss function* $\ell : \mathcal{Y} \times \mathcal{Y} \to [0, 1]$, where $\ell(y, z)$ is the loss of predicting $z$ when the true label is $y$. A learner $L : (\mathcal{X} \times \mathcal{Y})^* \to \mathcal{Y}^{\mathcal{X}}$ takes a training set $T = \{(x_1, y_1), \dots (x_n, y_n)\}$ consisting of a finite

number of iid samples from $\mu$ and outputs a classifier $L(T) \in \mathcal{Y}^{\mathcal{X}}$, where $\mathcal{Y}^{\mathcal{X}}$ is the set of functions from $\mathcal{X}$ to $\mathcal{Y}$. The loss of a classifier $c \in \mathcal{Y}^{\mathcal{X}}$ is $\ell_\mu(c) = \mathrm{E}_{(x,y) \sim \mu}[\ell(y, c(x))]$, and the expected loss of $L$ is $\ell_\mu(L) = \mathrm{E}_{T \sim \mu^n}[\ell_\mu(L(T))]$. Learning is with respect to a *concept class* $\mathcal{C} \subseteq \mathcal{Y}^{\mathcal{X}}$.

**Definition 1** (Agnostic learning of $\mathcal{C}$). *For countable[1] sets $\mathcal{X}, \mathcal{Y}$ and $\ell : \mathcal{Y} \times \mathcal{Y} \to [0, 1]$, learner $L$ agnostically learns $\mathcal{C} \subseteq \mathcal{Y}^{\mathcal{X}}$ if there exists a polynomial $p$ such that for any distribution $\mu$ over $\mathcal{X} \times \mathcal{Y}$ and for any $n \geq p(1/\epsilon, 1/\delta)$,*

$$\Pr_{T \sim \mu^n}\left[\ell_\mu(L(T)) \leq \min_{c \in \mathcal{C}} \ell_\mu(c) + \epsilon\right] \geq 1 - \delta.$$

*Further, $L$ and the classifier $L(T)$ must run in time polynomial in the length of their inputs.*

PAC learning refers to the special case when $\mu$ is additionally assumed to satisfy $\min_{c \in \mathcal{C}} \ell_\mu(c) = 0$. MUL, which is the focus of this paper, simply refers to case where $\mu$ is a *meta-distribution* over datasets $X \in \mathcal{X}$ and ground truth labelings $Y \in \mathcal{Y}$. We use capital letters to represent datasets as opposed to individual examples. A meta-classifier $c$ is a UL algorithm that takes an entire dataset $X$ as input and produces output, such as a clustering algorithm, $Z \in \mathcal{Y}$. As mentioned, true labels need only be observed for the training datasets – we may never observe the true labels of any problem encountered after deployment.[2] For a finite set $S$, let $\Pi(S)$ denote the set of clusterings or disjoint partitions of $S$ into two or more sets, e.g., $\Pi(\{1, 2, 3\})$ includes $\{\{1\}, \{2, 3\}\}$, i.e., the partition into clusters $\{1\}$ and $\{2, 3\}$. For a clustering $C$, denote by $\cup C = \cup_{S \in C} S$ the set of points clustered. Given two clusterings $Y, Z \in \Pi(S)$, the *Rand Index* $RI(Y, Z)$ measures the fraction of pairs of points on which they agree. *Adjusted Rand Index* (ARI) is a refined measure that attempts to correct RI by accounting for chance agreement [23]. We denote by $ARI(Y, Z)$ the adjusted rand index between two clusterings $Y$ and $Z$. We abuse notation and also write $ARI(Y, Z)$ when $Y$ is a vector of class labels, by converting it to a clustering with one cluster for each class label. We define the loss to be the fraction of pairs of points on which the clusterings disagree, assuming they are on the same set of points. If, for any reason the clusterings are not on the same set, the loss is defined to be 1, i.e.,

$$\ell(Y, Z) = \begin{cases} 1 - \mathrm{RI}(Y, Z) & if \cup Y = \cup Z \\ 1 & otherwise. \end{cases} \tag{1}$$

In *Euclidean clustering*, the points are Euclidean, so each dataset $X \subset \mathbb{R}^d$ for some $d \geq 1$. In meta-Euclidean-clustering, we instead aim to learn a clustering algorithm from several different training clustering problems (of potentially different dimensionalities $d$). Note that (Adjusted) Rand Index measures clustering quality with respect to an *extrinsic* ground truth. In many cases, such a ground truth is unavailable, and an *intrinsic* metric is useful. Such is the case, e.g., when choosing the number of clusters. We can compare and select from different clusterings of size $k = 2, 3, \ldots$ using the standard method of *Silhouette score* [24], which is defined for a Euclidean clustering as

$$\mathrm{sil}(C) = \frac{1}{|\cup C|} \sum_{x \in \cup C} \frac{b(x) - a(x)}{\max\{a(x), b(x)\}}, \tag{2}$$

where $a(x)$ denotes the average distance between point $x$ and other points in its own cluster and $b(x)$ denotes the average distance between $x$ and points in the closest alternative cluster.

## 3 Meta-unsupervised problems

The simplest approach to MUL is Empirical Risk Minimization (ERM), namely choosing from a family $\mathcal{U}$ any unsupervised algorithm with lowest empirical error on training set $T$, which we write as $\mathrm{ERM}_{\mathcal{U}}(T)$. The following lemma implies a logarithmic dependence on $|\mathcal{U}|$ and helps us solve several interesting MUL problems.

**Lemma 1.** *For any finite family $\mathcal{U}$ of UL algorithms, any distribution $\mu$ over problems $X, Y \in \mathcal{X} \times \mathcal{Y}$, and any $n \geq 1, \delta > 0$,*

$$\Pr_{T \sim \mu^n}\left[\ell_\mu(\mathrm{ERM}_{\mathcal{U}}(T)) \leq \min_{U \in \mathcal{U}} \ell_\mu(U) + \sqrt{\frac{2}{n} \log \frac{|\mathcal{U}|}{\delta}}\right] \geq 1 - \delta,$$

*where* $\mathrm{ERM}_{\mathcal{U}}(T) \in \arg \min_{U \in \mathcal{U}} \sum_{(X,Y) \in T} \ell(Y, U(X))$ *is any empirical loss minimizer over* $U \in \mathcal{U}$.

*Proof.* Fix $U_0 \in \arg \min_{U \in \mathcal{U}} \ell_\mu(U)$. Let

$$\epsilon = 2\sqrt{\frac{\log(1/\delta) + \log |\mathcal{U}|}{2n}} \ .$$

Invoking the Chernoff bound, we have

$$\Pr_{T \sim \mu^n} \left[ \frac{1}{n} \sum_{(X,Y) \in T} \ell(Y, U_0(X)) \geq \ell_\mu(U_0) + \epsilon/2 \right] \leq e^{-2n(\epsilon/2)^2}.$$

Define $S = \{U \in \mathcal{U} \mid \ell_\mu(U) \geq \ell_\mu(U_0) + \epsilon\}$. Applying the Chernoff bound, we have for each $U \in S$

$$\Pr_{T \sim \mu^n} \left[ \frac{1}{n} \sum_{(X,Y) \in T} \ell(Y, U(X)) \leq \ell_\mu(U) - \epsilon/2 \right] \leq e^{-2n(\epsilon/2)^2}.$$

In order for $\ell_\mu(\mathrm{ERM}_{\mathcal{U}}) \geq \min_{U \in \mathcal{U}} \ell_\mu(U) + \epsilon$ to happen, either some $U \in S$ must have empirical error at most $\ell_\mu(U) - \epsilon/2$ or the empirical error of $U_0$ must be at least $\ell_\mu(U_0) + \epsilon/2$. By the union bound, this happens with probability at most $|\mathcal{U}| e^{-2n(\epsilon/2)^2} = \delta$. $\square$

**Selecting the clustering algorithm/number of clusters.** Instead of the ad hoc parameter selection heuristics currently used in UL, MUL provides a principled data-driven alternative. Suppose one has $m$ candidate clustering algorithms, or parameter settings $C_1(X), \ldots, C_m(X)$ for each data set $X$. These may be derived from $m$ different clustering algorithms, or alternatively, they could represent the *meta-k* problem, i.e., how many clusters to choose from a single clustering algorithm where parameter $k \in \{2, \ldots, m+1\}$ determines the number of clusters. In this section, we show that choosing the right algorithm is essentially a multi-class classification problem given any set of problem meta-data features and cluster-specific features. Trivially, Lemma 1 implies that with $O(\log m)$ training problem sets one can select the $C_j$ that performs best across problems. For meta-$k$, however, this would mean choosing the same number of clusters to use across all problems, analogous to choosing the best single class for multi-class classification. To learn to choose the best $C_j$ on a problem-by-problem basis, suppose we have *problem features* $\phi(X) \in \Phi$ such as number of dimensions, number of points, domain (text/vision/etc.), and *cluster features* $\gamma(C_j(X)) \in \Gamma$ that might include number of clusters, mean distance to cluster center, and Silhouette score (eq. 2). Suppose we also have a family $\mathcal{F}$ of functions $f : \Phi \times \Gamma^m \to \{1, 2, \ldots, m\}$, which selects the clustering based on features (any multi-class classification family may be used for this purpose):

$$\arg \min_{f \in \mathcal{F}} \sum_i \ell \left( Y_i, C_{f(\phi(X_i), \gamma(C_1(X_i)), \ldots, \gamma(C_m(X_i)))}(X_i) \right).$$

The above $\mathrm{ERM}_{\mathcal{F}}$ is a reduction from the problem of selecting $C_j$ from $X$ to the problem of multi-class classification based on features $\phi(X)$ and $\gamma(C_1(X)), \ldots, \gamma(C_m(X))$ and loss as defined in eq. (1). As long as $\mathcal{F}$ can be parametrized by a fixed number of $b$-bit numbers, the ERM approach of choosing the "best" $f$ will be statistically efficient. If $\mathrm{ERM}_{\mathcal{F}}$ cannot be computed exactly within the time constraints, an approximate minimizer may be used.

**Fitting the threshold in single-linkage clustering**. To illustrate a concrete efficient algorithm, consider choosing the threshold parameter of a single linkage clustering algorithm. Fix the set of possible vertices $\mathcal{V}$. Take $\mathcal{X}$ to consist of undirected weighted graphs $X = (V, E, W)$ with vertices $V \subseteq \mathcal{V}$, edges $E \subseteq \{\{u, v\} \mid u, v \in V\}$ and non-negative weights $W : E \to \mathbb{R}_+$. The loss on clusterings $\mathcal{Y} = \Pi(\mathcal{V})$ is again as defined in Eq. (1). Note that Euclidean data could be transformed into the complete graph, e.g., with $W(\{x, x'\}) = \|x - x'\|$. Single-linkage clustering with parameter $r \geq 0$, $C_r(V, E, W)$ partitions the data such that $u, v \in V$ are in the same cluster if and only if there is a path from $u$ to $v$ such that the weight on each edge in the path is at most $r$. For generalization bounds, we simply assume that numbers are represented with a constant number of bits. The loss is defined as in (1). We have the following result.

**Theorem 1.** *The class $\{C_r \mid r > 0\}$ of single-linkage algorithms with threshold $r$ where numbers are represented using $b$ bits, can be agnostically learned. In particular, a quasilinear time algorithm achieves error $\leq \min_r \ell_\mu(C_r) + \sqrt{2(b + \log 1/\delta)/n}$, with prob. $\geq 1 - \delta$ over $n$ training problems.*

*Proof.* For generalization, we assume that numbers are represented using at most $b$ bits. By Lemma 1, we see that with $n$ training graphs and $|\{C_r\}| \leq 2^b$, we have that with probability $\geq 1 - \delta$, the error of ERM is within $\sqrt{2(b + \log 1/\delta)/n}$ of $\min_r \ell_\mu(C_r)$. It remains to show how one can find the best single-linkage parameter in quasilinear time. It is trivial to see that one can find the best cutoff for $r$ in polynomial time: for each weight $r$ in the set of edge weights across all graphs, compute the mean loss of $C_r$ across the training set. Since $C_r$ runs in polynomial time, loss can be computed in polynomial time, and the number of different possible cutoffs is bounded by the number of edge weights, which is polynomial in the input size. Thus the entire procedure takes polynomial time.

For a quasilinear time algorithm (in the input size $|T| = \Theta(\sum_i |V_i|^2)$), we run Kruskal's algorithm on the union graph of all the graphs in the training set (i.e., the number of nodes and edges are the sum of the number of nodes and edges in the training graphs, respectively). As the Kruskal's algorithm adds each new edge to its forest (in order of non-decreasing edge weight), effectively two clusters in some training graph $(V_i, E_i, W_i)$ have been merged. The change in loss of the resulting clustering can be computed from the loss of the previous clustering in time proportional to the product of the sizes of the two clusters that are being merged, since these are the only entities on which the clusterings, and thus the losses differ. Naïvely, this may seem to take $O(\sum_i |V_i|^3)$. However, note that, each pair of nodes begins separately and is updated, exactly once during the course of the algorithm, to be in the same cluster. Hence, the total number of updates is $O(\sum_i |V_i|^2)$. Since Kruskal's algorithm is quasilinear time itself, we deduce that the entire algorithm is quasilinear.

For correctness, it is easy to see that as the algorithm runs, $C_r$ has been computed for each possible $r$ at the step just preceding when Kruskal adds the first edge whose weight is greater than $r$. $\square$

**Outlier removal.** For simplicity, we consider learning a single hyperparameter pertaining to the fraction of examples, furthest from the mean, to remove. In particular, suppose training problems are classification instances, i.e., $X_i \in \mathbb{R}^{d_i \times m_i}$ and $Y_i \in \{1, 2, \ldots, k_i\}^{m_i}$. To be concrete, suppose one is using algorithm $C$ which is, say, K-means clustering. Choosing the parameter $\theta$ which is the fraction of outliers to ignore during fitting, one might define $C_\theta$ with parameter $\theta \in [0, 1)$ on data $x_1, \ldots, x_n \in \mathbb{R}^d$ as follows: (a) compute the data mean $\mu = \frac{1}{n}\sum_i x_i$, (b) set aside as outliers the $\theta$ fraction of examples where $x_i$ is furthest from $\mu$ in Euclidean distance, (c) cluster the data with outliers removed using $C$, and (d) assign each outlier to the nearest cluster center. We can trivially choose the best $\theta$ so as to optimize performance. With a single $b$-bit parameter $\theta$, Lemma 1 implies that this choice of $\theta$ will give a loss within $\sqrt{2(b + \log 1/\delta)/n}$ of the optimal $\theta$, with probability at least $1 - \delta$ over the sample of datasets. The number of $\theta$'s that need to be considered is at most the total number of inputs across problems, so the algorithm runs in polynomial time.

The meta-outlier-removal procedure can be extended directly to remove a different proportion of outliers from each test dataset. Conceptually, one may view the procedure as assigning weights to points in the test set of size $z$, where each of the $r$ outlier points (according to the learned threshold $\theta$) is assigned weight 0, and all the other points share an equal weight $1/(z - r)$. One could relax this hard assignment to instead have a distribution that re-adjusts the weights to assign a low but non-zero mass on the outlier points $r$. Then, these weights act as a prior in, e.g., the penalized and weighted K-means algorithm [25] that groups the points into clusters and a set of noisy or outlier points.

**Problem recycling.** For this model, suppose that each problem belongs to a set of common problem categories, e.g., digit recognition, sentiment analysis, image classification amongst the thousands of classes of ImageNet [26], etc. The idea is that one can recycle the solution to one version of the problem in a later incarnation. For instance, suppose that one trained a digit recognizer on a previous problem. For a new problem, the input may be encoded differently (e.g., different image size, different pixel ordering, different color representation), but there is a transformation $T$ that maps this problem into the same latent space as the previous problem so that the prior solution can be re-used. In particular, for each problem category $i = 1, 2, \ldots, N$, there is a latent problem space $\Lambda_i$ and a solver $S_i : \Lambda_i \to \mathcal{Y}_i$. Each problem $X, Y$ of this category can be transformed to $T(X) \in \Lambda_i$ with low solution loss $\ell(Y, S(T(X)))$. In addition to the solvers, one also requires a *meta-classifier* $M : \mathcal{X} \to \{1, 2, \ldots, N\}$ that, for a problem $X$, identifies which solver $i = M(X)$ to use. Finally, one has transformers $T_i : M^{-1}(i) \to \Lambda_i$ that map any $X$ such that $M(X) = i$ into latent space $\Lambda_i$. The output of the meta-classifier is simply $S_{M(X)}(T_{M(X)}(X))$. Lemma 1 implies that if one can optimize over meta-classifiers and the parameters of the meta-classifier are represented by $D$ $b$-bit numbers, then one achieves loss within $\epsilon$ of the best meta-classifier with $m = O\left(Db/\epsilon^2\right)$ problems.

# 4 The possibility of meta-clustering

In this section, we point out how the framing of meta-clustering circumvents Kleinberg's impossibility theorem [1] for clustering. To review, [1] considers clustering finite sets of points $X$ endowed with symmetric distance functions $d \in D(X)$, where the set of valid distance functions is:

$$D(X) = \{d : X \times X \to \mathbb{R} \mid \forall x, x' \in X, d(x, x') = d(x', x) \geq 0, d(x, x') = 0 \text{ iff } x = x'\}. \quad (3)$$

A clustering algorithm $A$ takes a distance function $d \in D(X)$ and returns a partition $A(d) \in \Pi(X)$. Kleinberg defines an axiomatic framework with the following three desirable properties, and proves no clustering algorithm $A$ can satisfy all of these properties. (**Scale-Invariance**) For any distance function $d$ and any $\alpha > 0$, $A(d) = A(\alpha \cdot d)$, where $\alpha \cdot d$ is the distance function $d$ scaled by $\alpha$. That is, the clustering should not change if the problem is scaled by a constant factor. (**Richness**) For any finite $X$ and clustering $C \in \Pi(X)$, there exists $d \in D(X)$ such that $A(d) = C$. Richness implies that for any partition there is an arrangement of points where that partition is the correct clustering. (**Consistency**) Let $d, d' \in D(X)$ such that $A(d) = C$, and for all $x, x' \in X$, if $x, x'$ are in the same cluster in $C$ then $d'(x, x') \leq d(x, x')$ while if $x, x'$ are in different clusters in $C$ then $d'(x, x') \geq d(x, x')$. The axiom demands $A(d') = A(d)$. That is, clusters should not change if the points within any cluster are pulled closer, and those in different clusters are pushed farther apart.

For intuition, consider clustering two points where there is a single distance. Should they be in a single cluster or two clusters? By richness, there must be some distances $\delta_1, \delta_2 > 0$ such that if $d(x_1, x_2) = \delta_1$ then they are in the same cluster while if $d(x_1, x_2) = \delta_2$ they are in different clusters. This, however violates scale-invariance, since the problems are at a scale $\alpha = \delta_2/\delta_1$ of each other. We show a natural meta-version of the axioms is satisfied by a simple meta-single-linkage clustering algorithm. The main insight is that prior problems can be used to define a scale in the meta-clustering framework. Suppose we define the clustering problem with respect to a non-empty training set of clustering problems. So a meta-clustering algorithm $M(d_1, C_1, \ldots, d_t, C_t) = A$ takes $t \geq 1$ training clustering problems with their ground-truth clusterings (on corresponding sets $X_i$, i.e., $d_i \in D(X_i)$ and $C_i \in \Pi(X_i)$) and outputs a clustering algorithm $A$. We can use these training clusterings to establish a scale. In particular, we will show a meta-clustering algorithm whose output $A$ always satisfies richness and consistency, and which satisfies the following variant of scale-invariance.

**Meta-scale-invariance** (MSI). Fix any distance functions $d_1, d_2, \ldots, d_t$ and ground truth clusterings $C_1, \ldots, C_t$ on sets $X_1, \ldots, X_t$. For any $\alpha > 0$, and any distance function $d$, if $M(d_1, C_1, \ldots, d_t, C_t) = A$ and $M(\alpha \cdot d_1, C_1, \ldots, \alpha \cdot d_t, C_t) = A'$, then $A(d) = A'(\alpha \cdot d)$.

**Theorem 2.** *There exists a meta-clustering algorithm that satisfies meta-scale-invariance and whose output always satisfies richness and consistency.*

*Proof.* There are a number of such clustering algorithms, but for simplicity we create one based on single-linkage clustering. Single-linkage clustering satisfies richness and consistency (see [1], Theorem 2.2). The question is how to choose its single-linkage parameter. With meta-clustering, the scale can be established using training data. One can choose it to be the minimum distance between any two points in different clusters across all training problems. It is easy to see that if one scales the training problems and $d$ by the same factor $\alpha$, the clusterings remain unchanged, and hence the meta-clustering algorithm satisfies meta-scale-invariance. □

# 5 Experiments

We conducted several experiments to substantiate the efficacy of the proposed framework under various unsupervised settings. We downloaded all classification datasets from OpenML (http://www.openml.org) that had at most 10,000 instances, 500 features, 10 classes, and no missing data to obtain a corpus of 339 datasets. We now describe in detail the results of our experiments.

**Selecting the number of clusters.** For the purposes of this section, we fix the clustering algorithm to be K-means and compare two approaches to choosing the number of clusters $k$, from $k = 2$ to 10. More generally, one could vary $k$ on, for instance, a logarithmic scale, or a combination of different scales. First, we consider a standard heuristic for the baseline choice of $k$: for each cluster size $k$ and each dataset, we generate 10 clusterings from different random starts for K-means and take one with best Silhouette score among the 10. Then, over the 9 different values of $k$, we choose the one

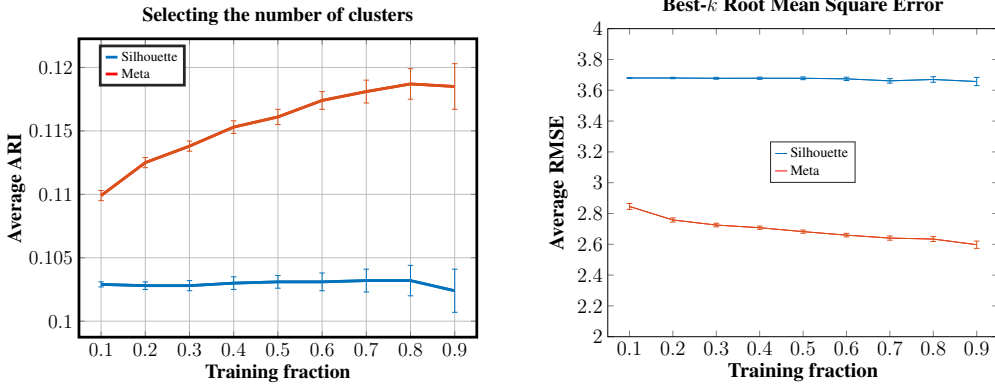

Figure 2: **(Left)** Average ARI scores of the meta-algorithm and the baseline for choosing the number of clusters, versus the fraction of problems used for training. We observe that the best $k$ predicted by the meta approach registered a much higher ARI than that by Silhouette score maximizer. **(Right)** The meta-algorithm achieved a much lower root-mean-square error than the Silhouette maximizer.

with greatest Silhouette score so that the resulting clustering is the one of greatest Silhouette score among all 90. In our meta-k approach, the meta-algorithm outputs $\hat{k}$ as a function of Silhouette score and $k$ by outputting the $\hat{k}$ with greatest estimated ARI. We evaluate on the same 90 clusterings for the 339 datasets as the baseline. To estimate ARI in this experiment, we used simple least-squares linear regression. In particular, for each $k \in \{2, \ldots, 9\}$, we fit ARI as a linear function of Silhouette scores using all the data from the meta-training set in the partition pertaining to $k$: each dataset in the meta-training set provided 10 target values, corresponding to different runs where number of clusters was fixed to $k$. As is standard, we define the best-fit $k_i^*$ for dataset $i$ to be the one that yielded maximum ARI score across the different runs, which is often different from $k_i$, the number of clusters in the ground truth (i.e., the number of class labels). We held out a fraction of the problems for test and used the remaining for training. We evaluated two quantities of interest: the ARI and the root-mean-square error (RMSE) between $\hat{k}$ and $k^*$. The meta-algorithm performed better than the baseline on both quantities (Fig. 2) (we performed 1000 splits to compute the confidence intervals).

**Selecting the clustering algorithm.** We consider the question which of given clustering algorithms to use to cluster a given data set. We illustrate the main ideas with $k = 2$ clusters. We run each of the algorithms on the repository and see which algorithm has the lowest average error. Error is calculated with respect to the ground truth labels by ARI (see Section 2). We compare algorithms on the 250 binary classification datasets with at most 2000 instances. The baselines are chosen to be five clustering algorithms from scikit-learn [27]: K-Means, Spectral, Agglomerative Single Linkage, Complete Linkage, and Ward, together with a second version of each in which each attribute is normalized to have zero mean and unit variance. Each algorithm is run with the default scikit-learn parameters. We implement the algorithm selection approach of Section 3, learning to choose a different algorithm for each problem based on problem and cluster-specific features. Given clustering $\Pi$ of $X \in \mathbb{R}^{d \times m}$, the feature vector $\Phi(X, \Pi)$ consists of the dimensionality, number of examples, minimum and maximum eigenvalues of covariance matrix, and silhouette score of the clustering $\Pi$:

$$\Phi(X, \Pi) = (d, m, \sigma_{\min}(\Sigma(X)), \sigma_{\max}(\Sigma(X)), \text{sil}(\Pi)),$$

where $\Sigma(X)$ denotes the covariance matrix of $X$, and $\sigma_{\min}(M)$ and $\sigma_{\max}(M)$ denote the minimum and maximum eigenvalues, respectively, of matrix $M$. Instead of choosing the clustering with best Silhouette score, which is a standard approach, the meta-clustering algorithm effectively learns terms that can correct for over- or under-estimates, e.g., learning for which problems the Silhouette heuristic tends to produce too many clusters. To choose which of the ten clustering algorithms on each problem, we fit ten estimators of accuracy by ARI based on these features. That is for each clustering algorithm $C_j$, we fit $\text{ARI}(Y_i, C_j(X_i))$ from features $\Phi(X_i, C_j(X_i)) \in \mathbb{R}^5$ over problems $X_i, Y_i$ using $\nu$-SVR regression, with default parameters as implemented by scikit-learn. Call this estimator $\hat{a}_j(X, C_j(X))$. To cluster a new dataset $X \in \mathbb{R}^{d \times m}$, the meta-algorithm then chooses $C_j(X)$ for the $j$ with greatest accuracy estimate $\hat{a}_j(X, C_j(X))$. The 250 problems were divided into train and test sets of varying sizes. The results, shown in Figure 3, demonstrate two interesting features. First, one can see that the

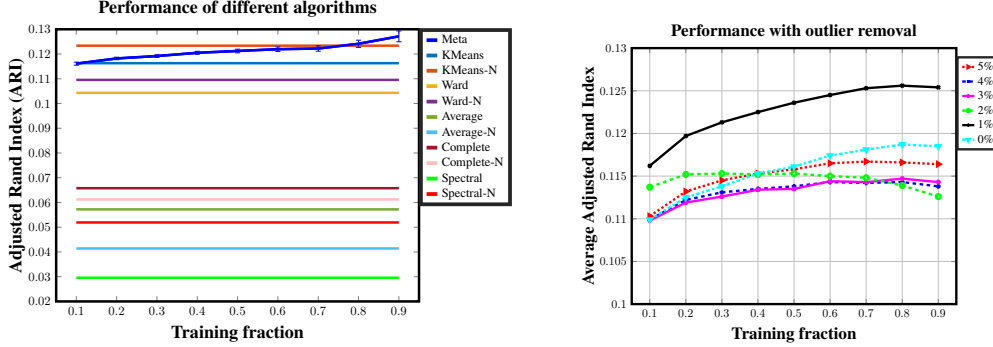

Figure 3: **(Left)** ARI scores of different clustering ($k$=2) algorithms on OpenML binary classification problems. The meta algorithm (95% confidence intervals are shown) is compared with standard clustering baselines on both the original data as well as the normalized data (denoted by "-N") . The meta-algorithm, given sufficient training problems, is able to outperform the best baseline algorithms by over 5%. Note that ARI is a strong measure of performance since it accounts for chance, and may even be negative. Moreover, the best possible value of ARI, i.e. by running the optimal algorithm for each dataset turned out to be about 0.16, which makes this improvement impressive. **(Right)** Outlier removal results are even better. Removing 1% of the instances as outliers improves the ARI score considerably. Interestingly, going beyond 1% decreases the performance to that without outlier removal. Our algorithm naturally figures out the right proportion (1%) in a principled way.

different baseline clustering algorithms had very different average performances, suggesting that a principled approach like ours to select algorithms can make a difference. Further, Figure 3 shows that the meta-algorithm, given sufficiently many training problems, is able to outperform, on average, all the baseline algorithms despite the fact the problems come from multiple heterogeneous domains.

**Removing outliers.** We also experimented to see if removing outliers improved average performance on the same 339 classification problems. Our objective was to choose a single best fraction to remove from all the meta-test sets. For each data set $X$, we removed a $p \in \{0, 0.01, 0.02, \ldots, 0.05\}$ fraction of examples with the highest euclidean norm in $X$ as outliers, and likewise for each meta-test set in the partition. We first clustered the data without outliers, and obtained the corresponding Silhouette scores. We then put back the outliers by assigning them to their nearest cluster center, and computed the ARI score thereof. Then, following an identical procedure to the meta-k algorithm of Section 5, we fitted regression models for ARI corresponding to complete data using the silhouette scores on pruned data, and measured the effect of outlier removal in terms of the true average ARI (corresponding to the best predicted ARI) over entire data. Again, we report the results averaged over 10 independent train/test partitions. As Fig. 3 shows, by treating 1% of the instances in each dataset as outliers, we achieved remarkable improvement in ARI scores relative to clustering with all the data as in section 5. As the fraction of data deemed outlier was increased beyond 2%, however, performance degraded. Clearly, we can learn what fraction to remove based on data, and improve the performance considerably even with such a simple algorithm.

**Deep learning binary similarity function.** In this section, we consider a new unsupervised problem of learning a binary similarity function (BSF) that predicts whether two examples from a given problem should belong to the same cluster (i.e., have the same class label). Formally, a problem is specified by a set $X$ of data and meta-features $\phi$. The goal is to learn a classifier $f(x, x', \phi) \in \{0, 1\}$ that takes two examples $x, x' \in X$ and the corresponding problem meta-features $\phi$, and predicts 1 if the input pair would belong to the same cluster (or have the same class labels). In our experiments, we take Euclidean data $X \subseteq \mathbb{R}^d$ (each problem may have different dimensionality $d$), and the meta-features $\phi = \Sigma(X)$ consist of the covariance matrix of the unlabeled data. We restricted our experiments to the 146 datasets with at most 1000 examples and 10 features, and formed disjoint *meta-training* and *meta-test* sets by randomly sampling pairs of examples from each dataset, and each resulting feature vector comprised of 55 covariance features, and 20 features from its pair. We assigned a label 1 to pairs formed by combining examples belonging to the same class, and 0 to those resulting from the different classes. Each dataset in the first category was used to sample data pairs for both the meta-training and the meta-internal_test (meta-IT) datasets, while the *second category*

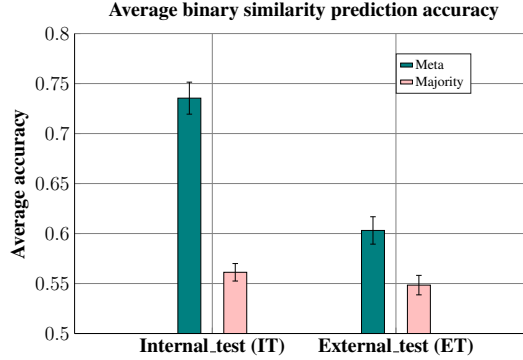

Figure 4: Mean accuracy and standard deviation on meta-IT and meta-ET data. Comparison between the fraction of pairs correctly predicted by the meta algorithm and the majority rule. Recall that meta-ET, unlike meta-IT, was generated from a partition that did not contribute any training data. Nonetheless, the meta approach dramatically improved upon the majority rule even on meta-ET.

*did not contribute any training data* and was exclusively used to generate only the meta-external_test (meta-ET) dataset. Our procedure ensured a disjoint intersection between the meta-training and the meta-IT data, and resulted in 10 separate (meta-training, meta-IT, meta-ET) triplets. Thus, we effectively turn small data into big data by combining examples from several small problems. The details of our sampling procedure, network architecture, and training are given in the Supplementary.

We tested our trained models on meta-IT and meta-ET data. We computed the predicted same class probability for each feature vector and the vector obtained by swapping its pair. We predicted the instances in the corresponding pair to be in the same cluster only if the average of these probabilities exceeded 0.5. We compared the meta approach to a hypothetical majority rule that had prescience about the class distribution. As the name suggests, the majority rule predicted all pairs to have the majority label, i.e., on a problem-by-problem basis we determined whether 1 (same class) or 0 (different class) was more accurate and gave the baseline the advantage of this knowledge for each problem, even though it normally would not be available at classification time. This information about the distribution of the labels was not accessible to our meta-algorithm. Fig. 4 shows the average fraction of similarity pairs correctly identified relative to the corresponding pairwise ground truth relations on the two test sets across 10 independent (meta-training, meta-IT, meta-ET) collections. Clearly, the meta approach greatly outperforms the majority rule on meta-IT, illustrating the benefits of the meta approach in a multi-task transductive setting. More interesting, still, is the significant improvement exhibited by the meta method on meta-ET, despite having its category precluded from contributing any data for training. The result clearly demonstrates the potential benefits of leveraging archived supervised data for informed decision making in unsupervised settings.

## Conclusion

Unsupervised settings are hard to define, evaluate, and work with due to lack of supervision. We remove the subjectivity inherent in them by reducing UL to supervised learning in a meta-setting. This helps us provide theoretically sound and practically efficient algorithms for questions like which clustering algorithm and how many clusters to choose for a particular task, how to fix the threshold in single linkage algorithms, and what fraction of data to discard as outliers. We also introduce the *meta-scale-invariance* (MSI) property, a natural alternative to scale invariance, and show how to design a single-linkage clustering algorithm that satisfies MSI, richness and consistency. Thus, we avoid Kleinberg's impossibility result and achieve provably good meta-clustering.

Finally, we automate learning of features across diverse data, and show how several small datasets may be effectively combined into big data that can be used for zero shot learning with neural nets. This is especially important due to two reasons: it (a) alleviates the primary limitation of deep nets that they are not suitable for extremely small datasets, and (b) goes beyond transfer learning with a few homogeneous datasets and shows how knowledge may be transferred from numerous heterogeneous domains to completely new datasets that contribute no data during training.

## Acknowledgments

We thank Lester Mackey for suggesting the title of this paper.

## Footnotes

[1]For simplicity of presentation, we assume that these sets are countable, but with appropriate measure theoretic assumptions the analysis in this paper can be extended to the infinite case.

[2]This differs from, say, online learning, where it is assumed that for each example, the ground truth is revealed once prediction is made.

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
