[Supplementary Material]

# A  Complete details: Deep learning a binary similarity function

We consider a new unsupervised problem of learning a binary similarity function (BSF) that predicts whether two examples from a given problem should belong to the same cluster (i.e., have the same class label). Formally, a problem is specified by a set $X$ of data and meta-features $\phi$. The goal is to learn a classifier $f(x, x', \phi) \in \{0, 1\}$ that takes two examples $x, x' \in X$ and the corresponding problem meta-features $\phi$, and predicts 1 if the input pair would belong to the same cluster (or have the same class labels).

In our experiments, we take Euclidean data $X \subseteq \mathbb{R}^d$ (each problem may have different dimensionality $d$), and the meta-features $\phi = \Sigma(X)$ consist of the covariance matrix of the unlabeled data. We restricted our experiments to the 146 datasets with at most 1000 examples and 10 features. We normalized each dataset to have zero mean and unit variance along every coordinate (hence, every diagonal element in the covariance matrix was set to 1).

We randomly sampled pairs of examples from each dataset to form *meta-training* and *meta-test* sets as described in the following section. For each pair, we concatenated the features to create data with 20 features (padding examples with fewer than 10 features with zeros). We then computed the empirical covariance matrix of the dataset, and vectorized the entries of the covariance matrix on and above the leading diagonal to obtain an additional 55 covariance features. All pairs sampled from the same dataset shared these covariance features, and thus we obtained a 75-dimensional feature vector per pair. Moreover, we derived a new binary label dataset in the following way. We assigned a label 1 to the pairs formed by combining examples belonging to the same class, and 0 to others.

### A.0.1  Sampling pairs to form meta-train and meta-test datasets

We formed a partition of the new datasets by randomly assigning each dataset to one of the two categories with equal probability. Each dataset in the first category was used to sample data pairs for both the meta-training and the meta-internal_test (meta-IT) datasets, while the second category did not contribute any training data and was exclusively used to generate only the meta-external_test (meta-ET) dataset.

We constructed meta-training pairs by randomly sampling pairs from each dataset in the first category. In order to mitigate the bias resulting from the variability in size of the different datasets, we restricted the number of pairs sampled from each dataset to at most 2500. Likewise, we obtained the meta-IT dataset by randomly sampling each dataset subject to at most 2500 pairs. Specifically, we randomly shuffled each dataset belonging to the first category, and used the first half (or 2500 examples, whichever was fewer) of the dataset for the meta-training data, and the following indices for the meta-IT data, again subject to maximum 2500 instances. This procedure ensured a disjoint intersection between the meta-training and the meta-IT data. Note that combining thousands of examples from each of hundreds of problems yields hundreds of thousands of examples, thereby turning small data into big data. This provides a means of making DNNs naturally applicable to data sets that might have otherwise been too small.

We created the meta-ET data using datasets belonging to the second category. Again, we sampled at most 2500 examples from each dataset in the second category. We emphasize that the datasets in the second category did not contribute any training data for our experiments. We performed 10 independent experiments to obtain multiple partitions of the datasets into two categories, and repeated the aforementioned procedure to prepare 10 separate (meta-training, meta-IT, meta-ET) triplets. This resulted in the following (average size +/- standard deviation) statistics for dataset sizes:

$$\text{meta-training and meta-IT} \quad : \quad 1.73 \times 10^5 \pm 1.07 \times 10^4$$
$$\text{meta-ET} \quad : \quad 1.73 \times 10^5 \pm 1.19 \times 10^4$$

In order to ensure symmetry of the binary similarity function, we introduced an additional meta-training pair for each meta-training pair in the meta-training set: in this new pair, we swapped the order of the feature vectors of the instances while replicating the covariance features of the underlying dataset that contributed the two instances (note that since the covariance features were symmetric, they carried over unchanged).

### A.0.2   Training neural models

For each meta-training set, we trained an independent deep net model with 4 hidden layers having 100, 50, 25, and 12 neurons respectively over just 10 epochs, and used batches of size 250 each. We updated the parameters of the model via the Adadelta [28] implementation of the stochastic gradient descent (SGD) procedure supplied with the Torch library[3] with the default setting of the parameters, specifically, interpolation parameter equal to 0.9 and no weight decay. We trained the model via the standard negative log-likelihood criterion (NLL). We employed ReLU non-linearity at each hidden layer but the last one, where we invoked the log-softmax function.

We tested our trained models on meta-IT and meta-ET data. For each feature vector in meta-IT (respectively meta-ET), we noted down the predicted same class probability. We added the predicted same class probability for the feature vector obtained with flipped order, as described earlier for the feature vectors in the meta-training set. We predicted the instances in the corresponding pair to be in the same cluster if the average of these two probabilities exceeded 0.5, otherwise we segregated them.

### A.0.3   Results

We compared the meta approach to a hypothetical majority rule that had prescience about the class distribution. As the name suggests, the majority rule predicted all pairs to have the majority label. That is, on each problem separately, we determined whether 1 (same class) or 0 (different class) was more accurate; and provided the baseline the advantage of this knowledge for each problem, even though it normally wouldn't be available at classification time. This information about the distribution of the labels was not made accessible to our meta-algorithm.

Fig. 4 shows the average fraction of similarity pairs correctly identified relative to the corresponding pairwise ground truth relations on the two test sets, and the corresponding standard deviations across the 10 independent (meta-training, meta-IT, meta-ET) collections. Clearly, the meta approach outperforms the majority rule on meta-IT, illustrating the benefits of the meta approach in a multi-task transductive setting. More interesting, still, is the significant improvement exhibited by the meta method on meta-ET, despite having its category precluded from contributing any data for training. The result clearly demonstrates the benefits of leveraging archived supervised data for informed decision making in unsupervised settings such as binary similarity prediction.

## Footnotes

[3]See `https://github.com/torch/torch7`.