[Reviews · NeurIPS 2018]

Reviewer 1



This paper introduces a meta-unsupervised learning (MUL) framework for unsupervised learning, where knowledge from heterogeneous supervised datasets is used to learn unsupervised learning tasks like clustering. The main idea is simple, but the authors define it in the perspective of agnostic learning and discuss its connections to related topics in a principled way. Using the empirical risk minimization, this paper demonstrate MUL on several unsupervised problems such as choosing the number of clusters in clustering, removing the outliers in clustering, and deep similarity learning. In particular, this paper shows that meta-clustering circumvents Kleinberg’s impossibility theorem by extending scale invariance to a meta-level. Experimental results show improvements over simple heuristic algorithms for the problems. + The proposed MUL framework is novel and interesting to my knowledge. + MUL is defined using agnostic learning and empirical risk minimization in a principled way. + Practical MUL instances are discussed and experimentally demonstrated. + The possibility theorem of meta-clustering is shown. - Only simple baseline algorithms are considered in experiments. I’ve enjoyed this paper and think MUL is interesting enough although the main idea can be viewed as a simple extension of meta-learning to unsupervised problems. This attempts may have been tried before. To my knowledge, however, the explicit formulation of MUL is novel and I don’t find any serious flaws in its analysis and discussions. Since I’m not an expert in this area, I would lower my confidence for now. But, I think this paper deserves publication. [Post-rebuttal] After reading other reviews as well as the rebuttal, I'm more confident that this paper is worth being presented in NIPS.

Reviewer 2



Summary: This paper uses meta-learning to solve unsupervised learning problem, especially the clustering problem. The author formulates the meta-unsupervised learning problem, and mentions several possible applications. In the experiments, the author shows that the proposed meta-unsupervised learning problem is helpful to solve some interesting clustering problem, such as deciding the number of clusters and selecting more useful clustering algorithms. strengths: 1. Theoretical justifications are nice. 2. The proposed application in deciding the number of clusters is useful. weaknesses: 1. Symbols are a little bit complicated and takes a lot of time to understand. 2. The author should probably focus more on the proposed problem and framework, instead of spending much space on the applications. 3. No conclusion section Generally I think this paper is good, but my main concern is the originality. If this paper appears a couple years ago, I would think that using meta-learning to solve problems is a creative idea. However, for now, there are many works using meta-learning to solve a variety of tasks, such as in active learning and reinforcement learning. Hence, this paper seems not very exciting. Nevertheless, deciding the number of clusters and selecting good clustering algorithms are still useful. Quality: 4 of 5 Clarity: 3 of 5 Originality: 2 of 5 Significance: 4 of 5 Typo: Line 240 & 257: Figure 5 should be Figure 3.

Reviewer 3



By considering a probability distribution over a family of supervised datasets, the authors propose to select a clustering algorithm from a finite family of algorithms or to choose the number of clusters among other tasks by solving a supervised learning problem that matches some features of the input dataset to the output dataset. For instance, in the case of selecting the number of clusters, they regress this number from a family of datasets learning a function that gives a "correct" number of clusters. The submission seems technically sound; the authors support the claim of the possibility of agnostic learning in two specific settings with a theoretical analysis: choosing an algorithm from a finite family of algorithms and choosing an algorithm from a family of single-linkage algorithms. Their framework also allows proposing an alternative to the desirable property of Scale-Invariance introduced by Kleinberg (2003) by letting the training datasets to establish a scale; this is translated into the Meta-Scale-Invariance desirable property. The authors then show that, with this version of the Scale-Invariance property, it is possible to learn a clustering algorithm that is also Consistent and Rich (as defined by Kleinberg (2003)). The work is clearly written, although it would be better to explain clearly the fact that now training samples are actually training datasets (lines 87 to 94). Also, I believe there is a better way to describe the concepts related to clustering (lines 94 to 114). Making a "Conclusion" section to summarize the results for people that want to understand the impact of the work quickly would be useful. To my knowledge, the idea of using meta-learning to propose an alternative to the Scale-Invariance desirable property to show that it is possible to achieve a good clustering is new and interesting. Moreover, the experimental results seem to back-up correctly the proposed claims which are already backed-up by a theoretical analysis which gives this work a considerable significance.